# Are nitrogen and carbon cycle processes impacted by common stream antibiotics? A comparative assessment of single vs. mixture exposures

Austin D. Gray[1,2¤]*, Emily Bernhardt[2]

1 Department of Biology, University of North Carolina at Greensboro, Greensboro, North Carolina, United States of America, 2 Department of Biology, Duke University, French Science Center, Durham, North Carolina, United States of America

¤ Current address: Department of Biological Science, Virginia Polytechnical Institute and State University, Blacksburg, Virginia, United States of America

* austindg@vt.edu

**Data Availability Statement:** Data for this project can be found on figshare using the links below https://figshare.com/articles/dataset/Gray_BGC_

## Abstract

A variety of antibiotics are ubiquitous in all freshwater ecosystems that receive wastewater. A wide variety of antibiotics have been developed to kill problematic bacteria and fungi through targeted application, and their use has contributed significantly to public health and livestock management. Unfortunately, a substantial fraction of the antibiotics applied to humans, pets and livestock end up in wastewater, and ultimately many of these chemicals enter freshwater ecosystems. The effect of adding chemicals that are intentionally designed to kill microbes, on freshwater microbial communities remains poorly understood. There are reasons to be concerned, as microbes play an essential role in nutrient uptake, carbon fixation and denitrification in freshwater ecosystems. Chemicals that reduce or alter freshwater microbial communities might reduce their capacity to degrade the excess nutrients and organic matter that characterize wastewater. We performed a laboratory experiment in which we exposed microbial community from unexposed stream sediments to three commonly detected antibiotics found in urban wastewater and urban streams (sulfamethoxazole, danofloxacin, and erythromycin). We assessed how the form and concentration of inorganic nitrogen, microbial carbon, and nitrogen cycling processes changed in response to environmentally relevant doses (10 μg/L) of each of these antibiotics individually and in combination. We expected to find that all antibiotics suppressed rates of microbial mineralization and nitrogen transformations and we anticipated that this suppression of microbial activity would be greatest in the combined treatment. Contrary to our expectations we measured few significant changes in microbially mediated functions in response to our experimental antibiotic dosing. We found no difference in functional gene abundance of key nitrogen cycling genes $nosZ$, $mcrA$, $nirK$, and $amoA$ genes, and we measured no treatment effects on $NO_3^-$ uptake or $N_2O$, $N_2$, $CH_4$, $CO_2$ production over the course of our seven-day experiment. In the mixture treatment, we measured significant increases in $NH_4^+$ concentrations over the first 24 hours of the experiment, which were indistinguishable from controls

study_Raw_Data_2021_xlsx/17008964/1 https://doi.org/10.6084/m9.figshare.17009126.v1.

**Funding:** AG Society of Environmental Toxicology and Chemistry Student Training Exchange Opportunity Award or (STEO) https://awards.setac.org/student-training-exchange-opportunity/ The funders had no role in study design, data collection and analysis, decision to publish, or preparation of the manuscript. This research was also supported by North Carolina Sea Grant.

**Competing interests:** The authors declare that no competing interests exist.

within six hours. Our results suggest remarkable community resistance to pressure antibiotic exposure poses on naïve stream sediments.

## Introduction

Pharmaceuticals are classified as "new emerging pollutants", due to their detection in almost all environmental matrices at increasing concentrations and no regulations regarding their release into the environment [1–3]. Organisms in urban streams are exposed to a suite of chemicals via wastewater and receive among the highest diversity of pharmaceuticals, which may be sourced from septic fields, leaky sewage infrastructure, combined sewer overflows, wastewater treatment plant effluent, and pet waste [4–9]. Of the various types of pharmaceuticals antibiotics are the most frequently used and detected in aquatic environments [2, 10].

Antibiotics may compound the effects of excess nutrient loading and more frequent scouring flows that are known to reduce the capacity for urban stream ecosystems to support biodiversity and to sequester or transform excess nutrients [5, 11, 12]. Antibiotics are manufactured to inhibit or kill microbes at the point of use, but because they continue to be in their active form in wastewater, they can have unintended negative consequences in receiving ecosystems [13]. Microbes are principally responsible for the transformation and assimilation of excess nitrogen (N) in urban streams (e.g., denitrification, nitrogen fixation, and organic matter decomposition) [14–16]. If antibiotics alter benthic microbial communities, this may result in significant changes to these critical N processing function. Excessive reactive N in streams can cause eutrophication, harmful algal blooms, oxygen depletion, and acidification [17–20]. Recent experimental evidence has demonstrated that pharmaceuticals can suppress processes such as primary production and respiration [21–23]. There has been little experimental work examining the effects of pharmaceuticals on N cycling, but it has been shown that the antibiotic sulfamethoxazole can inhibit denitrification [24, 25].

In previous work, we found that sulfamethoxazole, danofloxacin, and erythromycin were the most commonly detected antibiotics in streams in the Piedmont of North Carolina [7, 26, 27]. These commonly detected antibiotics each represent a different class and mode of action regarding bacterial death or inhibition. Erythromycin and sulfamethoxazole are both broad spectrum antibiotics that are widely used for human and veterinary purposes. Both compounds inhibit growth but do so through different mechanisms. Erythromycin is a macrolide antibiotic that irreversibly binds to 50S ribosomal subunits [28], while sulfamethoxazole is a sulfonamide that inhibits nucleic acid, protein synthesis, and cell wall permeability [29]. Danofloxacin is a synthetic fluoroquinolone that is primarily used in veterinary medicine for the treatment of respiratory disease. Danofloxacin inhibits DNA replication by deactivation of bacterial DNA gyrase and topoisomerase IV [30]. Prior studies in the same region measured low rates of N removal and high methane concentrations in streams with detectable concentrations of these three antibiotics [31, 32].

Studies investigating the effects emerging contaminants have on ecosystem function lag behind other well-defined drivers of global environmental change [13]. The objective of this study was to assess how stream sediment biogeochemistry and microbial activity would change in response to dosing with relevant concentrations of these three common antibiotic pollutants. In our experiment, each antibiotic was added alone or in combination to a sediment slurry. We measured changes in the concentration of $NH_4^+$, $NO_3^-$, $N_2$, $N_2O$, $CH_4$, and $CO_2$ over the course of 7 days (24 hours for $NH_4^+$ and $NO_3^-$) following the experimental dosing

**Table 1. Site data for surface water and sediment used in the study.**

| Site | Lat | Long | pH | DO (mg/L) | Conductivity (µS/cm) | Temperature (°C) | $NH_4^+$ (µg N $L^{-1}$) | $NO_3^-$ (µg N $L^{-1}$) | % Organic Matter |
|---|---|---|---|---|---|---|---|---|---|
| Lake Brandt | 36.173262 | -79.8379 | 8.7 | 4.8 | 555 | 23°C | 18.2 | 116 | 1.2 |

using destructive sampling of replicates at multiple time steps. All replicates were fully oxygenated at the start of the experiment, and all slurries were hypoxic/anoxic within 24 hours, allowing us to examine the impact of our antibiotic dosing under fluctuating redox conditions. We expected to find that all antibiotic treatments would lead to declines in the assimilation and transformation of one or more forms of inorganic N and organic C relative to the controls. We anticipated measuring the greatest number and magnitude of impacts in our antibiotic mixture treatment, predicting that the effects of the three antibiotics would be antagonistic, additive, or synergistic due to their divergent modes of action.

## Materials and methods

### Sample collections

Sediment and surface water used in microcosm construction were collected from a forested stream near Lake Brandt in Greensboro, NC, USA (36.173262, -79.83790) (Table 1). Previous surveys of this site found no antibiotic residues in compartments sampled. Surface water was collected in 5 L acid-washed carboys, and sediment was collected on-site and transferred into sterile Ziploc® plastic bags. At the sampling site, pH, dissolved oxygen, and conductivity were recorded. Once transported to the lab, surface water was filtered using 0.7 µm GF/F filters. Sediments were homogenized and stored on ice in the dark until microcosm construction. Surface water samples were stored in a refrigerator at 4°C. In the laboratory, ash-fry dry mass and sediment particle size classification were determined from subsamples (Table 1; S1 File). Subsamples of surface water was analyzed on a Lachat QuickChem 8500 automated system (Lachat Instruments, Loveland, Colorado, USA) to determine background concentrations of $NH_4^+$ and $NO_3^-$ (Table 1).

### Antibiotic solutions

Sulfamethoxazole (SMX), danofloxacin (DAN), and erythromycin (ETM) (all > 98% purity) were obtained from Sigma-Aldrich (St. Louis, MO) (Table 2). Stock solution (1 mg/L) was added to filtered surface water to yield an initial concentration of 10 µg/L for each antibiotic treatment. Mixture treatments consisted of each antibiotic having an initial concentration of 10 µg/L.

### Microcosm construction

Microcosms consisted of 200 mL glass serum bottles. Bottles contained 100 g of wet naïve sediments and 100 mL of filtered surface water. Control treatments contained no spiked antibiotic solution. Microcosms were sealed with gas impermeable butyl rubber stoppers and crimped

**Table 2. Antibiotic information of those used in the present study.**

| Antibiotic | Class | Mode of Action | CAS No. | pKa | LogKow |
|---|---|---|---|---|---|
| Sulfamethoxazole (SMX) | Sulfonamide | Inhibiting nucleic acids, protein synthesis, and folic acid synthesis | 723-46-6 | 1.6, 5.7 | 0.89 |
| Danofloxacin (DAN) | Fluoroquinolone | Inhibition of bacterial DNA replication | 112398-08-0 | 6.22, 9.43 | 0.44 |
| Erythromycin (ETM) | Macrolide | Prevents growth by binding irreversibly to 50S ribosomal subunits | 114-07-8 | 7.7, 8.9 | 1.6–3.1 |

aluminum seals (Geo-Microbial Technologies). All microcosms were spiked with a nutrient enrichment to reduce substrate limitations. Enrichments consisted of glucose (0.2 g C), sodium acetate (0.12 g C), and ammonium chloride (0.16 mg N). Enrichments were mixed with filtered stream water prior to the addition of antibiotics. The treatments (control, SMX, DAN, ETM, and SMX+DAN+ETM mixture) were further divided into separate oxic and hypoxic or anoxic treatments. After antibiotics solutions (10 μg/L) were spiked into the respective microcosm and sealed, microcosms were shaken. Samples collected for the oxic treatments consisted of 3 replicates per treatment per time point (n = 12 per antibiotic treatment), where the same microcosm was sampled repeatedly with samples collected at 0, 6, 12, and 24 h.

Microcosms were allowed to go hypoxic or anoxic naturally without amending or purging them with $N_2$. Preliminary sediment oxygen demand studies conducted in the lab showed dissolved oxygen levels reached 0.5 mg/L by 24 h (S2 File). The oxygen demand study concluded at 24 h, while samples taken during the hypoxic or anoxic period were collected past this time. Calculated oxygen saturation levels from 0 to 24 h were $\geq 5.5\%$. Following 24 h, we suspect that oxygen levels were $\leq 5.5\%$. It is important to note that in homogeneous sediments, methanogen activity and denitrification can be confined to anoxic microsites within sediments [33]. Due to this, even in the presence of oxygen, these processes can still occur. Microcosms designated to become hypoxic or anoxic naturally were sampled over 7 days, with sampling occurring on days 2, 4, and 7. Each treatment consisted of five replicates per time point (n = 25). Due to the nature of sampling, serum bottles were destructively sampled with five randomly selected bottles at each sampling point per treatment (n = 125). All microcosms were stored in the dark to limit photodegradation of antibiotics throughout the study period. Antibiotic concentration in water or sediment were not measured at the conclusion of the study.

## $NH_4^+$ and $NO_3^-$ measurements

$NH_4^+$ and $NO_3^-$ concentrations were analyzed over 24 h. At each specified sampling point (0, 6, 12, and 24 h), 10 mL of water was collected. Water was collected from the same bottle at each time point and filtered through ashed GF/F filters. Water was not replaced as it might alter microcosm conditions at subsequent sampling periods (increased DO, dilution). Samples were then kept in a -20˚C freezer until analysis. At the conclusion of the study, samples were analyzed on the Lachat QuickChem 8500 automated system (Lachat Instruments, Loveland, Colorado, USA) to determine $NH_4^+$ and $NO_3^-$ concentrations. $NH_4^+$ and $NO_3^-$ concentrations were calculated over time to determine rate of consumption over 24 h (μg N $L^{-1}$ hour.).

## $N_2O$, $CH_4$, and $CO_2$ measurements

Greenhouse gas or GHG ($N_2O$, $CH_4$, $CO_2$) concentrations were measured over 7 days, with sampling times on day 2, 4, and 7. Each sample was collected from the gaseous headspace following shaking the bottles to release any gas trapped within sediment. 10 mL of headspace gas was extracted and transferred to a 9 mL glass vial. Glass vials containing headspace gas were purged with $N_2$ and evacuated prior to use. Gas samples were stored inverted in the dark until sample analysis. $N_2O$, $CH_4$, and $CO_2$ concentrations were measured using a Teledyne Tekmar 7000 headspace autosampler (Teledyne Tekmar, Mason, Ohio, USA) to inject samples into a Shimadzu GC-17A ver.3 gas chromatograph with a Porapak Q column and electron capture detector [34]. Concentrations were acquired from NIST grade calibration standards. The linearity of the calibration was determined from the $R^2 > 0.97$. Concentrations of $N_2O$, $CH_4$, and $CO_2$ were normalized to the dry weight of sediment (g) (nmol $g^{-1}$ DW). Dry-weight concentrations were plotted over time for each microcosm to estimate the rate of gas production (nmol $g^{-1}$ DW $day^{-1}$).

## N$_2$ measurements

N$_2$ was measured using a Membrane Inlet Mass Spectrometer or MIMS to determine dissolved N$_2$ and Ar concentrations in surface water overlaying sediment. Serum bottles (n = 25/time point) were destructively sampled due to the use of the MIM probe. Standards for N$_2$ concentrations (humid-atmosphere-equilibrated deionized water stirring in high-precision water baths) at 22˚C and 24˚C were run every six samples. Standards were run in triplicate along with samples collected at day 2,4, and 7. N$_2$ concentrations were achieved using the MIMS_-gasfunction package in R version 3.5.2 (R Core Team 2017). N$_2$ concentrations were converted from mg/L to µmol/g and normalized to dry weight of sediment (g). Dry-weight concentrations were plotted over time for each microcosm to estimate the rate of N$_2$ production (µmol g$^{-1}$ DW day$^{-1}$).

## Functional gene abundance

We measured the abundance of key genes necessary for biochemical processes evaluated in the present study. We measured 16s rRNA, *nosZ* (encodes for nitrous oxide reductase), *mcrA* (encodes the alpha subunit of the methyl-coenzyme M reductase (MCR), which catalyzes the last step in methanogenesis), *nirK* (encodes for the copper-containing nitrite reductase), and *amoA* (encodes for ammonia monooxygenase) [34–36] (Table 3). At the conclusion of the study, subsamples of sediment were stored in -20˚C freezer in LifeGuard Soil Preservation Solution (Qiagen) until extraction. We extracted DNA from sediments in triplicate with PowerSoil DNA Isolation kits (MoBio Laboratories, Carlsbad, California, USA). DNA was quantified by a Nanodrop Spectrophotometer (Thermo Scientific) at an absorbance of 260 nm. Following extraction, samples were diluted to 3 ng/mL. Genes of interest DNA were amplified using primers purchased from Integrated DNA Technologies (Coralville, Iowa, USA) (Table 3) and iTaq Universal SYBR mix (Bio-Rad Laboratories, Hercules, California, USA). The average efficiency of the qPCR reaction ranged from 81% to 111%, all standard curves had R$^2$ values $\geq$0.98 (Table 3). Copies were normalized to the amount of extracted sediment.

## Statistical analysis

Due to non-normal distribution throughout the dataset, non-parametric analyses were conducted. Outlier analysis was performed in Microsoft Excel to determine the upper and lower bounds. The upper and lower bounds were calculated using the following equations:

Q3 + (1.5 * IQR) and Q1 - (1.5 * IQR)

**Table 3. Primers used in the functional gene abundance qPCR assay.**

| Gene | Forward Sequence | Reverse Sequence | Product size (bp) | Efficiency | Reference |
|---|---|---|---|---|---|
| **16s rRNA** | 5'-TAA CCT GGG AAC GCG ATT T-3' | 5'- CCA CTA CCC TCT ACC ACA CT-3' | 55 | 111% | [37] |
| ***nosZ*** | 5'- AGA ACG ACC AGC TGA TCG ACA-3" | 5'- TCC ATG GTG ACG CCG TGG TTG-3' | 300 | 101% | [38] |
| ***mcrA*** | 5'- AAA GTG CGG AGC AAT CAC C-3' | 5'TCG TCC CAT TCC TGC TGC ATT GC-3' | 186 | 87% | [39] |
| ***nirK*** | 5'-TCA TGG TGC CGC GGA CGG-3' | 5- GAA CTT GCC GGT GCC CAG AC-3' | 326 | 88% | [40] |
| ***amoA*** | 5'- GGG GTT TCT ACT GGT GGT-3' | 5-' CCC CTC BGS AAA VCC TTC TTC-3' | 491 | 81% | [41] |

Q1 = Quartile 1

Q3 = Quartile 3

IQR = Interquartile Range

For each respective assay, a Kruskal-Wallace test compared experimental treatments change in concentration to the control to determine differences. For rate analysis, $NH_4^+$ and $NO_3^-$ concentrations were plotted over time (24 hours) for each microcosm (µg N $L^{-1}$ hour). $N_2O$, $N_2$, $CH_4$, and $CO_2$ rates were assessed by plotting concertation across time (7 days) for each microcosm (nmol $g^{-1}$ DW $d^{-1}$ or µmol $g^{-1}$ DW $d^{-1}$). A Pairwise Wilcox test was run to determine distinct differences in significant treatments. Gene copy numbers were compared using Kruskal-Wallace test to determine differences among treatments. Correlation analysis was conducted between gene copy number and respective products and reactants. Data were analyzed using R Studio (3.5.2) and graphs were made in GraphPad Prism (8.5.1).

## Results

### Oxic period

**$NH_4^+$ and $NO_3^-$.**  $NH_4^+$ and $NO_3^-$ concentrations from our collection site near Lake Brandt were 18.2 and 116 µg N $L^{-1}$, respectively (Table 1). We spiked our microcosms with 1 mg/L of $NH_4Cl$ or 0.16 mg N. At the start of the experiment $NH_4^+$ concentrations ranged between 388 to 673 µg N $L^{-1}$ ± 49.12 ($\bar{x}$ ± 95%CI) and $NO_3^-$ concentrations ranged from 166 to 281 µg N $L^{-1}$ ± 16.67 ($\bar{x}$ ± 95%CI) across all treatments (Fig 1A, 1C; S3 File). Mean DAN $NH_4^+$ concentration at 0 h were exceptionally high (1330 ±178 µg N $L^{-1}$) ($\bar{x}$ ± SE) compared to the other treatments and were identified as statistical outliers (Fig 1A), so rate calculations for this treatment were only calculated at the second sampling point.

In the controls, $NH_4^+$ concentration declined over a 24 h period from 459 ± 46 to 33 ± 9.4 µg N $L^{-1}$ $h^{-1}$ ($\bar{x}$ ± SE) (Fig 1A), with a rate of depletion of -19.2 ± 1.6 µg N $L^{-1}$ $h^{-1}$ ($\bar{x}$ ± SE). There was no detectable difference between the control, DAN, and ETM treatments, while $NH_4^+$ concentration remain unchanged in the SMX treatment (0.13 ± 1.6 µg N $L^{-1}$ $h^-$1) ($\bar{x}$ ± SE), but increased significantly in the mixture treatment (11.5 ± 0.7 µg N $L^{-1}$ $h^{-1}$) ($\bar{x}$ ± SE), over the course of the 24 h oxic assay (Fig 1A, 1B; Table 4).

Mean $NO_3^-$ concentration from the control treatment (166 ± 9) ($\bar{x}$ ± SE), was 1.4 to 4.3 times lower than single and mixture antibiotic treatments at 0 h (p<0.05: Fig 1C; S3 File). By 24 h, $NO_3^-$ concentrations from each treatment had declined compared to concentrations at 0 h (Fig 1C). The rate of decline in $NO_3^-$ concentration ranged from -4.7 to -6.7 µg N $L^{-1}$ $h^{-1}$ (Fig 1D; Table 4) where no treatments differed from the control treatment (p>0.05; Fig 1D; Table 4).

### Hypoxic or anoxic period

**$N_2O$.**  Our initial recordings of $N_2O$ concentrations 0.009 to 0.014 nmol $g^{-1}$ DW ± 0.001 ($\bar{x}$ ±95%CI) at day 2 found there was no difference among treatments when compared to the control (Fig 2A; S4 File). This trend was consistent at day 4 and 7 where there was no difference, despite each treatment declining in concentration (p>0.05; Fig 2A; S4 File). Control rate of N2O production was not significantly different than the other treatments used in the study with the mean rate of $N_2O$ production ranging from -0.0004 to -0.001 nmol $g^{-1}$ DW $d^{-1}$ (p>0.05 Fig 2B; Table 4).

$N_2$ concentrations (16.8 to 20 µmol $g^{-1}$ DW ± 0.65) ($\bar{x}$ ±95%CI) measured at day 2 were not significantly different among treatment (p>0.05; Fig 2C; S4 File). At day 4, control and mixture treatment concentrations were 1.2 to 1.5 times lower than the SMX, DAN, and ETM

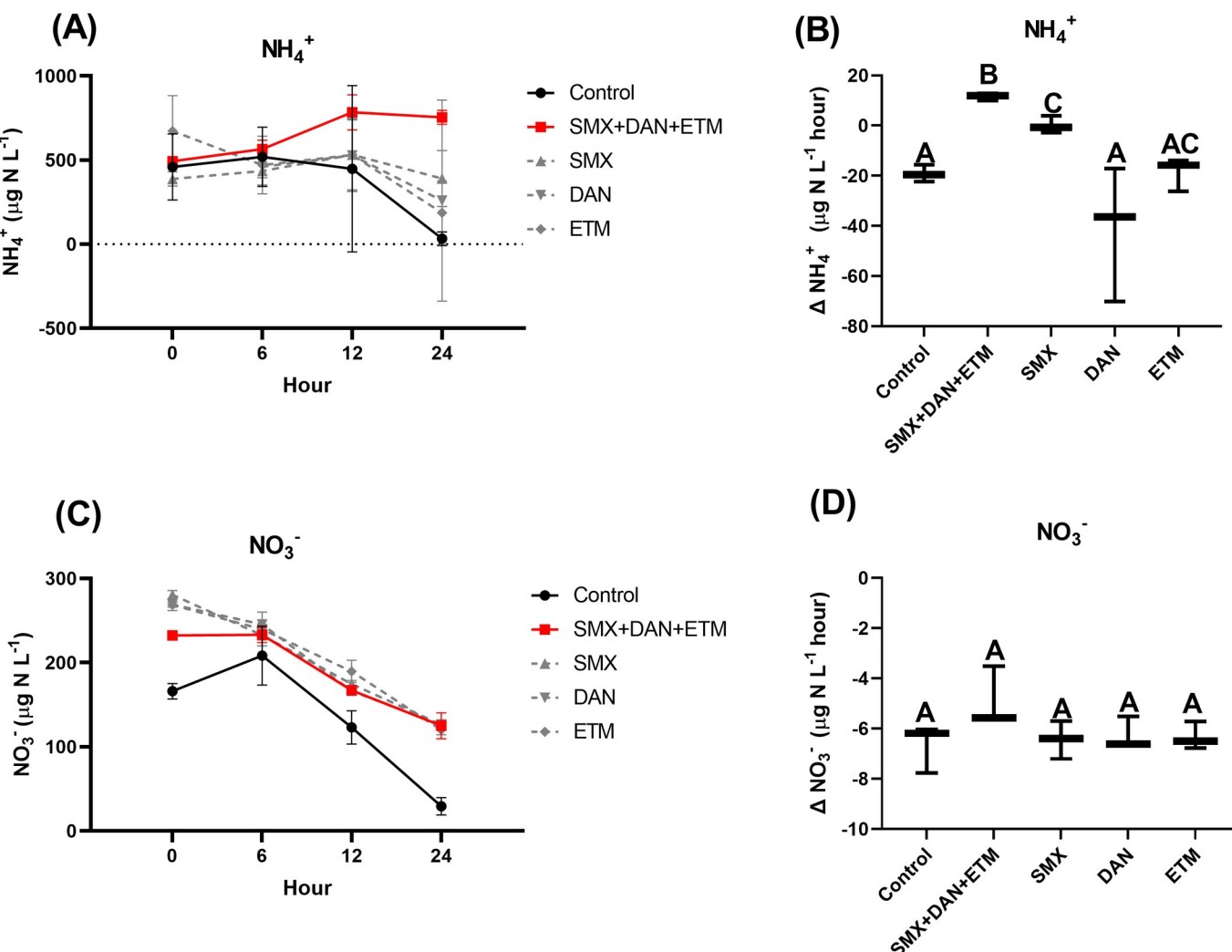

**Fig 1.** Mean ± SE dissolved (A) $NH_4^+$, (C) $NO_3^-$ concentrations in microcosms plotted across time (0,6,12, and 24 hour) and whisker plots (min to maximum) of the change in (B) $NH_4^+$ and (D) $NO_3^-$ concentration from microcosms exposed to control and experimental treatments. Identical letters above bars (1B and 1D) indicate treatments that are not significantly different as determined by post hoc Pairwise Wilcox Test, while different letters indicate rates that are significantly different.

**Table 4. Mean ± SE consumption and production rates of nitrogen species (NH4+, NO3-, N₂, N₂O), CH₄, and CO₂.**

| | $\mu g\ N\ L^{-1}\ h^{-1}$ | | $\mu mol\ g^{-1}\ DW\ d^{-1}$ | $nmol\ g^{-1}\ DW\ d^{-1}$ | | |
| --- | --- | --- | --- | --- | --- | --- |
| | $NH_4^+$ | $NO_3^-$ | $N_2$ | $N_2O$ | $CH_4$ | $CO_2$ |
| **Control** | -19.2 ± 1.6 | -6.7 ± 0.5 | -0.05 ± 0.21 | -0.001 ± 0.0002 | 6.3 ± 0.8 | 20.4 ± 24.0 |
| **SMX+DAN+ETM** | 11.5 ± 0.7* | -4.9 ± 0.6 | 0.31 ± 0.25 | -0.001 ± 0.0001 | 3.3 ± 1.0 | 66.0 ± 2.9 |
| **SMX** | 0.13 ± 1.6* | -6.4 ± 0.4 | -0.45 ± 0.12 | -0.0004 ± 0.0001 | 3.7 ± 0.6 | 68.4 ± 11.4 |
| **DAN** | -12.8 ± 7.7 | -6.3 ± 0.3 | -0.36 ± 0.60 | -0.001 ± 0.0003 | 6.4 ± 1.3 | 47.5 ± 23.7 |
| **ETM** | -18.65 ± 3.1 | -6.3 ± 0.3 | 0.28 ± 0.44 | -0.001 ± 0.0003 | 7.5 ± 1.6 | 40.4 ± 14.8 |

*denotes treatments that were significantly different than control treatment.

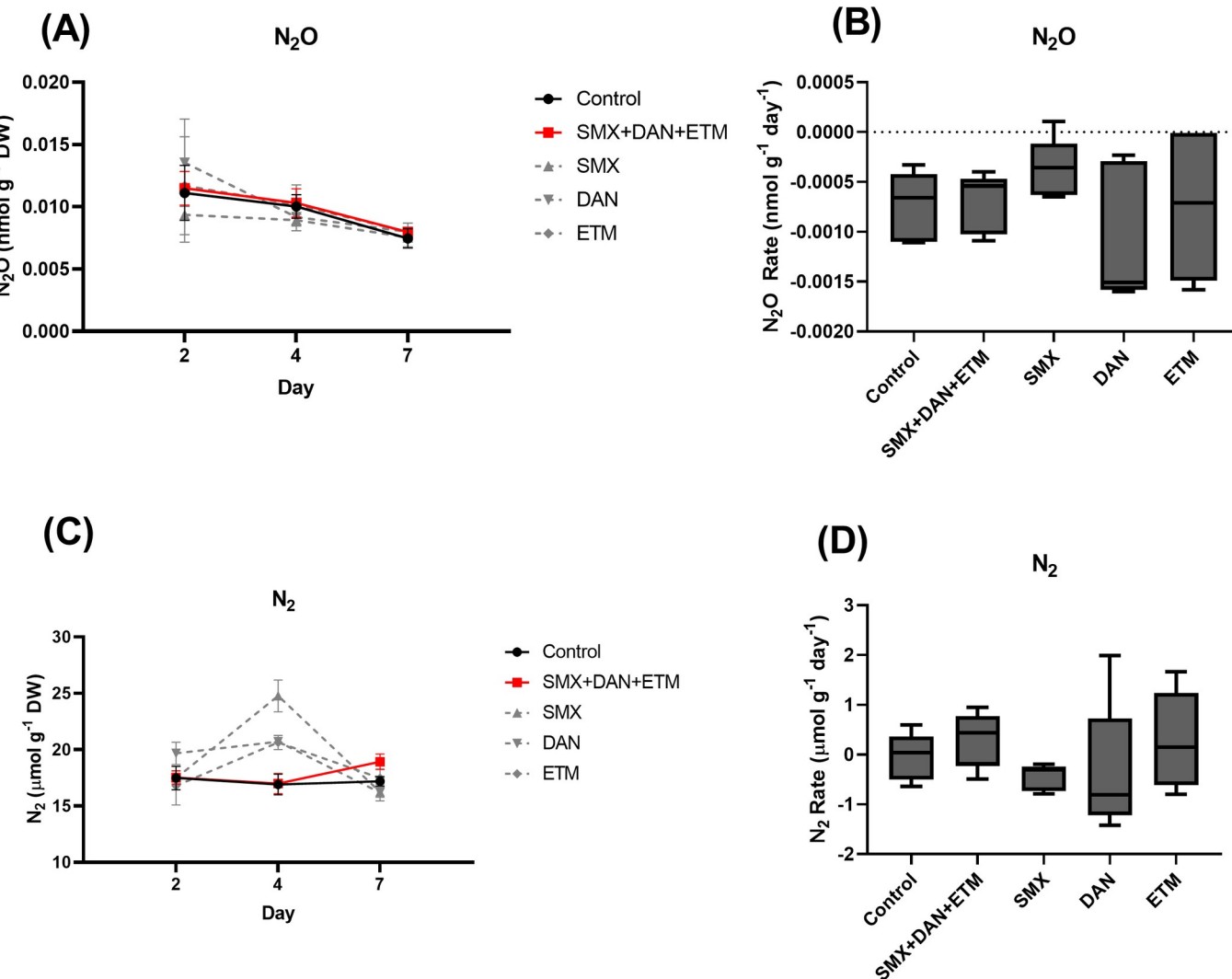

**Fig 2.** Mean ± SE dissolved (A) $N_2O$, (C) $N_2$ concentrations in microcosms plotted across time (Day 2, 4, and 7) and whisker plots (min to maximum) of the rate of production for (B) $N_2O$ and (D) $N_2$ from microcosms exposed to control and experimental treatments.

treatments ($p < 0.05$; S4 File; Fig 2C). By day 7, $N_2$ concentrations were not significantly different among treatments ($p > 0.05$; Fig 2C, S4 File). The rate of $N_2$ production ranged between -0.05 to 0.31 μmol $g^{-1}$ DW $d^{-1}$, where we observed no significant difference among treatments. (Fig 2D; S4 File).

**$CO_2$.** Mean $CO_2$ concentrations at day 2 ranged from 322 to 592 nmol $g^{-1}$ DW ± 171.8 ($\bar{x}$ ±95% CI) (Fig 3A; S4 File). Control, SMX, and DAN treatments concentration on day 2 were 1.6 to 1.8 times higher than the SMX and mixture treatment ($p < 0.05$; Fig 3A; S4 File). At day 4, we saw that the mean control $CO_2$ concentrations were now 1.4 times lower than the mixture treatment but this reduction was not significant ($p > 0.05$; Fig 3A; S4 File). At day 7, we observed no significant differences in $CO_2$ concentrations among treatments ($p > 0.05$; Fig 3A). The increase in $CO_2$ on day 7 compared to concentrations on day 2 suggests that as the microcosms became hypoxic or anoxic, $CO_2$ production was promoted. Rate of $CO_2$ production ranged from 20.4 to 68.4 nmol $g^{-1}$ DW $d^{-1}$ across treatments, with no difference across treatments ($p > 0.05$; Fig 3B; Table 4).

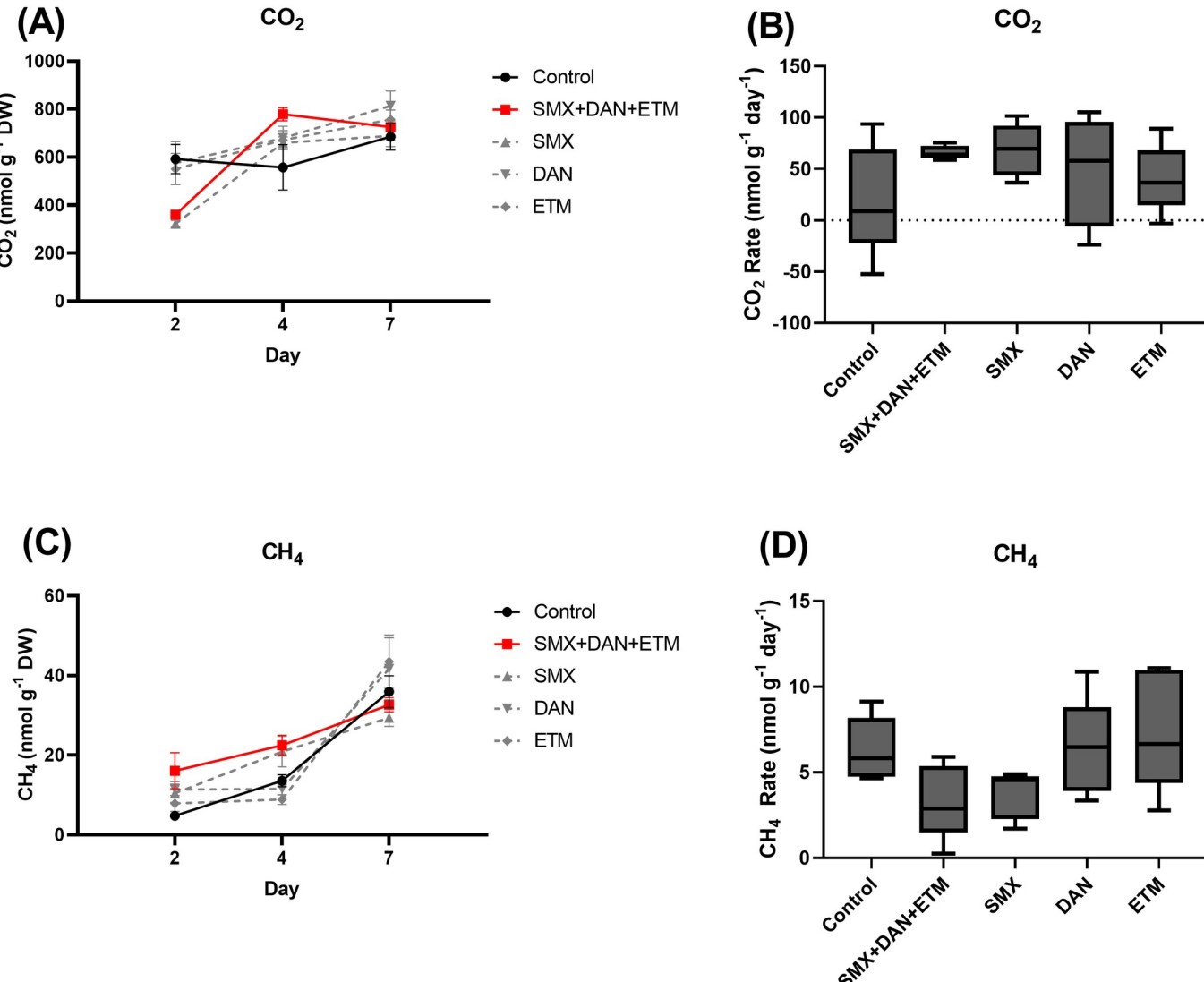

**Fig 3.** Mean ± SE dissolved (A) $CH_4$, (C) $CO_2$ concentrations in microcosms plotted across time (Day 2, 4, and 7) and whisker plots (min to maximum) of the rate of production for (B) $CH_4$ and (D) $CO_2$ from microcosms exposed to control and experimental treatments.

**$CH_4$.** Mean $CH_4$ concentrations at day 2 ranged from 4.8 to 16.1 nmol $g^{-1}$ DW ± 3.24 ($\bar{x}$ ±95%CI) (Fig 3C; S4 File). Control treatment concentration at day 2 (4.8 ± 0.5 nmol $g^{-1}$ DW) was 1.6 to 3.4 times higher than the single and mixture antibiotic treatments ($p<0.05$; Fig 3A; S4 File). At day 4, control, DAN, and ETM treatment concentrations were 1.7 to 2.5 times lower than then SMX and mixture treatments ($p<0.05$; Fig 3C; S4 File). At the conclusion of the study on day 7, there was no difference in $CH_4$ concentrations across treatments (Fig 3C; S4 File). Over the 7-day assay, $CH_4$ concentrations increased in all treatments at rates ranging from 3.3 to 7.5 nmol $g^{-1}$ DW $d^{-1}$, with no significant difference across treatments compared to the controls ($p>0.05$; Table 4; Fig 3D).

**Functional gene abundance.** Mean number of copies of respective genes ranged from 1.86E+05 to 1.56E+06 copies per gram of sediment (S5 File). We observed no difference in copy number abundances for each respective gene. Correlation analysis revealed there was no

significant correlation between functional genes and the rates of consumption or production respective genes are responsible for.

## Discussion

In our assessment we initially expected to find that all antibiotic treatments would lead to declines in the assimilation and transformation of one or more forms of inorganic N and organic C relative to the controls. We anticipated the greatest impact to be seen in the mixture treatment due to the unknown interactions of these compounds possibly resulting in an additive or synergistic effect. However, at the conclusion of the present study we found that there were generally no differences pertaining to the impact antibiotics (both single and mixture exposures) posed on nutrient assimilation and transformation compared to control or antibiotic free treatments. The only assessment where we saw the mixture treatment behave differently than the control was in the rates of $NH_4^+$ uptake in our oxic assay.

Ecological assessments that include contaminant mixtures are highly underrepresented in environmental science. Although mixtures are a more realistic representative of the natural environment, the mechanisms behind their combined impact is complex due to varying modes of actions whose interaction is not widely understood. Chen et al. [42] and González-Pleiter et al. [43] both reported that mixtures of contaminants (pesticides and antibiotics), can result in a synergistic growth inhibition on photosynthetic aquatic organisms. The mixture used in the present study represented different antibiotic classes that each have a unique action mechanism (Table 2). The increase in $NH_4^+$ as a result of exposure to antibiotic mixtures is a topic that warrants further discussion.

We saw no change in $NH_4^+$ concentration in the SMX treatment during the oxic assay, which could be the result of SMX either inhibiting nitrification or enhancing mineralization. Nitrification inhibition by sulfonamides, such as SMX has been previously reported by [44]. The difference in antibiotic effect on nitrification can be influenced by whether it is a broad or narrow-spectrum antibiotic [45], where narrow spectrum can kill or inhibit a specific species, as opposed to broad spectrum that can target both gram-negative and positive bacteria (nitrifiers are gram-positive). However, each antibiotic used in the study was a broad-spectrum antibiotic, suggesting that effects on nitrification are not consistent or universal across all broad-spectrum drugs. We observed no effect of DAN or ETM treatments on $NH_4^+$ uptake when compared to the control treatment. However, the effects of the DAN+ETM+SMX treatment was distinct from the SMX alone treatment, indicating the potential for a synergistic effect for one or both antibiotics in combination with SMX.

The most interesting result from this experiment was the significant increase in $NH_4^+$ from the mixture treatment. Exposure to antibiotics can affect community functions such as mineralization [46]. These findings from the mixture treatment suggest both potentially enhanced mineralization and reduced nitrification as factors contributing to the increase in $NH_4^+$ over time. We cannot distinguish between these two possible mechanisms but our results suggest future work should focus on how antibiotic exposure may influence these two processes.

Based on $NO_3^-$ consumption along with $N_2O$ and $N_2$ production rates, it does not appear that antibiotic treatments, whether single or mixture, had any negative effect on denitrification. Underwood et al. [25] showed that SMX altered or inhibited denitrification at a concentration of 1.2 μg/L, a concentration almost 10 times lower than what was used in the present study. The different outcomes are likely a result of different experimental designs/exposures. Underwood et al. [25] used bacterial enrichment cultures from groundwater samples and exposed the isolated denitrifying community to SMX with no sediment. The impact of SMX and the other antibiotic treatments used in the present study may be less pronounced due to

the presence of sediment and/or the microbial populations that colonized sediment, potentially neutralizing antibiotics. SMX is made up of weak forces that favor desorption rather than sorption [47]. Antibiotics that bind to sediment can result in reduced bioavailability and potency, making them less effective at inhibiting bacterial community growth [48, 49]. While the reduced bioavailability may be a factor here, sediment used in the study had relatively low percent organic matter content (Table 1). It is possible that low sorption capabilities resulted in microbes being less effected by antibiotics.

The lack of differences observed may have been influenced by microbes within each microcosm acclimating or proving to be resistant to antibiotics [50]. By resistance we mean that for a majority of our endpoints measured, they were not altered as a result of antibiotic exposure. It is also important to note that the minimum inhibitory concentrations (MIC) of SMX, ETM, and DAN range between 0.25 to 16 μg/ml which is the equivalent of 250 to 16000 μg/L, concentrations higher than what was used in the present study [51–53]. Due to utilizing concentrations below MICs reported in literature, findings may be the result of concentrations not being high enough to produce a varying effect from the control (excluding SMX and mixture impact on $NH_4^+$). It is worth noting that antibiotic resistance occurs naturally in nature [54]. The sediment bacterial communities may have already had resistance that resulted in antibiotics having no effect on nitrification (excluding SMX and mixture), denitrification, and methanogenesis.

Rossi et al. [55] reported that fluctuations in gene expression can inform the future outcomes of a variety of cellular states. When functional gene abundance was evaluated, we observed no difference in gene copy numbers of the functional genes across treatments, supporting our previous claims that acclimation or microbial resistance occurred. This finding is consistent with findings from [56], where when multiple studies assessed the relationship between gene abundance and end-products, only 38% showed that the concentration of products or reactants correlated with gene abundance. Although levels used in the present study are below MICs previously reported, bacterial responses to antibiotics can be ununiformed and vary across times [55], where as a result of exposure, communities that persist initial exposures can express genes higher to compensate for those potentially lost. Rocca et al. [56] recommended using metagenomic assessments of protein encoding genes as viable step in future studies. However, since our study found that end products and gene expression did not differ across treatments, utilizing metatranscriptomics in follow up work would be a proactive approach, allowing for a functional profile to be compiled based on genes expressed that are specific to biogeochemical processes of interest.

This study's novelty was the paired evaluation of mixture exposure toxicity alongside single exposure toxicity in naïve stream sediments. Similar work has been conducted to in marine and wetland sediments, although streams are underreported [50, 57]. It is also important to note that these types of assessments rarely utilize environmentally relevant concentrations [50, 58]. Instead, previous work in the literature has focused on using therapeutic doses (mg/L or mg/kg) [59]. Demonstrating differences in mixture compared to single exposures are necessary as they more accurately reflect the natural environment. The mixture effect on nitrification is particularly interesting due to its importance in N cycling. In the present study we were able to address the question regarding how stream microbes respond to continuous exposure to antibiotics. However, stream organisms are exposed to both continuous and episodic exposures, largely influenced by runoff and wastewater effluent release [60]. Episodic or pulse exposures may affect toxicokinetic processes that result in varying results than what was found in the present study due to fluctuating concentrations that may be below or higher than MICs depending on design and relevance to study areas. Evaluating these exposure pathways would also provide more evidence into whether microbial communities and resistant to antibiotic exposure in certain study areas.

Nitrification of $NH_4^+$ to $NO_3^-$ is an environmental concern due to excess $NO_3^-$ leaching into groundwater (potential drinking water hazard), eutrophication, and its acute toxicity associated with wildlife. Any inhibition or reduction of the nitrification process, as demonstrated by the SMX and mixture treatments, can be seen as a positive regarding regulating $NO_3^-$ in freshwaters. The potential of enhanced mineralization suggest that naïve stream sediments exposed to mixtures may experience greater bioavailability of nutrients within the system. The antibiotics used here are simply a small representation of the contaminants present in stream ecosystems. Moving forward it would be of interest to evaluate more emerging contaminants to determine how or whether nitrification is altered.

Freshwater systems globally are plagued with various synthetic chemicals entering at varying rates. Mixture assessments are vastly underrepresented and as the present study shows, in the case of nitrification, its effect can differ when compared to results from single exposures. Although there were not many differences observed between single and mixtures, future work into these dynamics are warranted due to the varying rates of introduction and concentrations of synthetic chemicals providing pressure on microbial communities. Stream microbes from the present study appear to be resistant to antibiotic pressure in some cases. However, there may be sublethal effects aside from alterations to biogeochemical cycling that are not fully understood and require further investigation.

## Supporting information

**S1 File. Sediment information.** Sediment particle size composition of stream sediment.
(PDF)

**S2 File. Sediment oxygen demand.** Sediment oxygen levels from ambient and nutrient enrichment.
(PDF)

**S3 File. NH4+ and NO3- concentrations.** Mean ± SE $NH_4^+$ and $NO_3^-$ concentrations at each sampling point.
(PDF)

**S4 File. N₂O, N₂, CH₄, CO₂ concentrations.** Mean ± SE concentrations of $N_2O$, $N_2$, $CH_4$, and $CO_2$ from each experimental treatment over the 7-day study.
(PDF)

**S5 File. Functional gene abundance.** Mean gene copy number per gram of sediment for each functional gene investigated in the study.
(PDF)

## Acknowledgments

The authors would like to that Brooke Hassett for her assistance in this project along with the members of the Bernhardt Lab and Duke River Center at Duke University. We would also like to thank the UNCG Biology Department for allowing us to utilize the Cutter Laboratory, Dr. Kasie Raymann for providing space for qPCR sample preparation, and Dr. Joseph Santin for his guidance.

## Author Contributions

**Conceptualization:** Austin D. Gray, Emily Bernhardt.

**Data curation:** Austin D. Gray.

**Formal analysis:** Austin D. Gray.

**Funding acquisition:** Austin D. Gray.

**Investigation:** Austin D. Gray.

**Methodology:** Austin D. Gray, Emily Bernhardt.

**Resources:** Emily Bernhardt.

**Software:** Emily Bernhardt.

**Validation:** Austin D. Gray.

**Visualization:** Austin D. Gray.

**Writing – original draft:** Austin D. Gray.

**Writing – review & editing:** Austin D. Gray, Emily Bernhardt.

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
