## [Decision Letter · Decision Letter 0]

8 Nov 2021

PONE-D-21-30040Are Nitrogen and Carbon Cycle Processes Impacted by Common Stream Antibiotics? A Comparative Assessment of Single vs. Mixture ExposuresPLOS ONE

Dear Dr. Gray,

Thank you for submitting your manuscript to PLOS ONE. After careful consideration, we feel that it has merit but does not fully meet PLOS ONE’s publication criteria as it currently stands. Therefore, we invite you to submit a revised version of the manuscript that addresses the points raised during the review process.

We look forward to receiving your revised manuscript.

Kind regards,

John J. Kelly

Academic Editor

PLOS ONE

“AG

SETAC Student Exchange Training Opportunity

https://awards.setac.org/student-training-exchange-opportunity/

No”

Reviewers' comments:

Reviewer's Responses to Questions

**Comments to the Author**

1. Is the manuscript technically sound, and do the data support the conclusions?

Reviewer #1: Yes

Reviewer #2: Yes

2. Has the statistical analysis been performed appropriately and rigorously? 

Reviewer #1: Yes

Reviewer #2: Yes

3. Have the authors made all data underlying the findings in their manuscript fully available?

Reviewer #1: No

Reviewer #2: No

4. Is the manuscript presented in an intelligible fashion and written in standard English?

Reviewer #1: Yes

Reviewer #2: Yes

5. Review Comments to the Author

Reviewer #1: I have reviewed the manuscript “Are nitrogen and carbon cycle processes impact by common stream antibiotics? A comparative assessment of single vs. mixture exposures” submitted for consideration of publication at PLOS One. The authors report results of a stream sediment incubation study assessing impacts of sulfamethoxazole, danofloxacin, and erythromycin singly and in combination on microbiological activities. Only ammonium uptake was affected, in that it was reduced with sulfamethoxazole, and switched to production with the combination of all 3 compounds. The manuscript is generally well written and easy to read, however there are several typos and misplaced or missing commas that should be paid close attention to on the final edit. I request few modifications to the content, however I do request more substantive discussion on the synergistic mechanisms at play in the ammonium release results, and more thought on cellular processes when discussing lack of differences or processes more likely to be affected. Please consider all the comments as detailed below in a revision.

Title: “impacted” instead of “impact”

L85: “irreversibly bind”

L87: “is a synthetic”

L89: delete “enzyme of”; “Prior environmental studies”

L115: “In the laboratory, ash-free dry mass”

L143: is this the correct formatting for Figure reference?

L179: MIMS

L227: “spiked”

L238 and all Figures: There are no post-hoc letters on the figures in my downloaded pdf. Please make sure these come through; it’s harder to interpret without them.

L296: note in text “copies per gram sediment”

L305-306: the wording on this statement needs to be fixed, and clarified

L327: “in” instead of “from”.

L327-332: The consideration of additive vs. synergistic effects in the discussion was intriguing. Please revisit that here. I expect the authors don’t want to get too speculative but there is an observation about reduced uptake vs. cell death that is possible. It would be even more interesting to do a back-of the envelope calculation of how many bacteria would have died to produce this much ammonium over a day….

L335 and L337: Start these sentences with Underwood et al. [26] instead of a number

L341: sorption of organic compounds is likely a major mechanism underlying apparent resistance or resilience, please add more emphasis and consideration here. Is the resistance/resilience biological, or only apparent due to physical buffering – would there be a threshold where chronic effects manifest?

L344: please don’t imply that evolutionary processes were happening over a day or a week (unless that is what is meant and if so needs to be substantiated); delete “adapting to or”, replace with “acclimating” if desired

L344 and L345: Sentences should not begin with “This”: specify what the subject is.

L344: Re lack of differences: are sulfamethoxazole, danofloxacin, and erythromycin anaerobe-effective? Also, antibiotic resistance exists in plenty of populations even in “naïve” sediments: many bacteria compete with one another, like all organisms.

L346-348: This is true but it would be more interesting to elaborate on the cellular mechanism: Unless cells are killed and the DNA decomposed, there will be no decline in gene copies; but in the short term cellular functions including reproduction should be inhibited. In fact, the same would apply to metagenomic assessment so I don’t agree with the viability of metagenomics as a next step as posed in L349. Considering which bacterial populations are fastest-growing and which biogeochemical functions they support might be more interesting in this paragraph.

L373-374: This sentence could benefit from some grammatical attention (who are “they”- and there are two different implied subjects, one in each question listed, neither subject is stated directly) and also some elaboration on what is meant by “cost”. Please improve this statement.

Supplemental Table 3: There is a lingering comment in the document, please fix.

Authors are required to make raw data publicly available, if I understand correctly, and I do not see a link to data repository in the manuscript.

Reviewer #2: This manuscript describes the results of an experiment where a stream sediment microbial community with little prior exposure to antibiotics was exposed to low levels of antibiotics. The authors were seeking to understand more about how antibiotics given to humans and animals and subsequently released into wastewater and then receiving streams could impact the ecosystem-related functions of microorganisms in those streams. Laboratory microcosms were set up to expose the sediment to one of three antibiotics or to a mixture of the antibiotics. Nutrients also were added to the microcosms. The chemical and biological components of the microcosms were interrogated with numerous chemical and biological measurements over the course of a 7-day incubation. It was predicted that the antibiotics would have a significant reduction in the capability of the sediment community to conduct nitrogen transformations and uptake organic carbon. Ultimately the authors found that the antibiotics had fairly little impact on the functioning of these microbial communities. Nitrification may have been impacted during the oxic phase of the incubations (<24 hours) by the sulfonamide and mixed antibiotic treatments, but it was suggested further study would be needed to clarify this impact.

Overall, I found this to be a well conducted study and a cohesive manuscript. The results indicated that low levels of antibiotics over a short duration may not have a big impact on the nitrogen cycling or organic C uptake by the sediment microbial community in streams. This information will be useful in future work that addresses impacts of additional antibiotics/pharmaceuticals or varying concentrations of these substances on aquatic microorganisms and/or the ecosystem services provided by these systems. I do not have any major criticisms of the work, but I do have some minor points that need clarification and questions about some of the results interpretation presented in the discussion.

Also, please see the details regarding PLOS ONE data availability. A researchgate account is listed as the place where data is housed, but I did not see the underlying data upon going to that account. PLOS asks that all data used to calculate means, etc. are available in their raw count form. Please make these data available as supplemental info or in another data repository.

Specific comments

Throughout the manuscript there are some minor grammar issues – e.g. wrong tense or extra words to delete. The authors should look for and fix these small details; here are a few examples:

L85 – should be “binds” not “binding”

L95 – should be “common” not “commonly”

L227 – should be “spiked” not “spike”

L232 – delete “used”

L305 – should be “there” instead of “the”

L108 – What is meant by “for the assay” in this sentence? Is this the water that was collected for all the experimental setups? Please clarify.

L109-110 – Please cite this work, if possible, even if it is from a non-peer reviewed source. It would be useful to have the data showing that antibiotics are undetectable at this site – then the reader could evaluate what antibiotics were assayed and what method was used, etc.

L191 – This is semantics, but by convention it is written as the “16S rRNA gene” and is not italicized. Italics are used for protein-coding genes.

L312 – I agree that antibiotic mixtures are a more realistic scenario experienced by aquatic microbes, but how are the mechanisms behind their impact more complex than a single antibiotic exposure? Is there evidence that mixtures create a larger impact than would be predicted based on responses to individual antibiotics? Further support for this statement is needed or it needs to be presented as an opinion.

L314 – 318 – Could another alternative be that the concentration of antibiotic added was not at a level that would impact many microbes? What is the minimum inhibitory concentration for these antibiotics and how does this relate to the concentration used? Have other observed effects on microbial activity at these concentrations. If not, then it is possible that the dose was too low to impact process. I agree that low concentrations create a more realistic scenario, and the ones used here matched measured levels in other waterways, so it was a good target.

Perhaps a more continuous dosing of antibiotics, which could be brought by wastewater discharge would have a larger effect at this low concentration than a single pulse of the antibiotic. I believe additional discussion related to the antibiotic concentration used would aid in interpretation of the results.

Also in this section – in what evidence is there that these communities are highly resilient? My question isn’t whether the communities are resilient – this is a hypothesis, but more how are the authors defining that term. I tend to define it as the ability to recover from disturbance to a pre-disturbance state. This would indicate that at some point there was a big change in microbial community following antibiotic addition, but it recovered quickly. To me that data don’t support that as being a primary conclusion. I see it more that the community was resistant to this disturbance – i.e. they were not impacted or that there was enough redundancy in the community that it superseded any losses. A more specific indication of what is meant here by resilient would clarify what hypotheses the authors are considering in the interpretation of their data.

L317 – 318 – Why not discuss this here then? What do you think it means? Or what do you mean exactly by this statement?

L323 – In what way does being a broad or narrow spectrum antibiotic matter to nitrification? Be more specific in this statement.

L327 – come back to this any additional thoughts?

L340 - 342 – Clarify this. I think I agree with your point but make clear the distinction between the studies. Did the previous study also use sediment? Sediment inclusion could make a big difference as diffusion of the antibiotic into the sediment may not be that high in 7 days.

The wording, “…antibiotics exposed within sediment…” is a bit confusing. Is this stating that the antibiotics were added to the sediment or that the microbes in the sediment rather than the water were exposed to the antibiotics?

L343 – More explanation is needed in this paragraph? I do not follow the logic that the communities changed in structure, but this led to a lack of difference in measured gene copies. If the community composition is changing rapidly, I would expect the nitrogen-related gene concentrations to change rapidly as well, as only a few select taxa are capable of many of the relevant processes.

I agree with the reference cited that the concentration of a particular gene often does not relate to the activity levels at any given moment for that gene product. To get at this idea, gene expression (mRNA or protein copies) would need to be quantified. However, here the end products (N2 gas, etc.) were measured and they were similar between the control and treatment, so I would not expect that gene expression levels changed much either. I would suggest metatranscriptomics rather than metagenomics would be a better measure – find out which organisms are actually responding to the different microcosm setups.

L373 – 374 – The wording of this sentence is not clear. Please re-phrase. The negative, “…not how they are impacted…” make it difficult to interpret. Also, what is meant by “cost”? Is there a cost to the microbes for resisting antibiotics?

6. PLOS authors have the option to publish the peer review history of their article (what does this mean?). If published, this will include your full peer review and any attached files.

Reviewer #1: No

Reviewer #2: No

---

## [Author Response · Author response to Decision Letter 0]

2 Dec 2021

Review Comments to the Author

The authors appreciate the thorough review of our submitted manuscript titled “Are nitrogen and carbon cycle processes impacted by common stream antibiotics? A comparative assessment of single vs. mixture exposures”. Below we have responded to specific reviewers’ comments and have included information as to where specific changes were made in the revised manuscript (identified by line number). We also adjusted the reference list to support information for in text revisions. 

Reviewer 1 Overall comments on the manuscript

Reviewer #1: I have reviewed the manuscript “Are nitrogen and carbon cycle processes impacted by common stream antibiotics? A comparative assessment of single vs. mixture exposures submitted for consideration of publication at PLOS One. The authors report results of a stream sediment incubation study assessing impacts of sulfamethoxazole, danofloxacin, and erythromycin singly and in combination on microbiological activities. Only ammonium uptake was affected, in that it was reduced with sulfamethoxazole, and switched to production with the combination of all 3 compounds. The manuscript is generally well written and easy to read, however there are several typos and misplaced or missing commas that should be paid close attention to on the final edit. I request few modifications to the content; however, I do request more substantive discussion on the synergistic mechanisms at play in the ammonium release results, and more thought on cellular processes when discussing lack of differences or processes more likely to be affected. Please consider all the comments as detailed below in a revision.

Reviewer 2 overall comments on manuscript

Reviewer #2: This manuscript describes the results of an experiment where a stream sediment microbial community with little prior exposure to antibiotics was exposed to low levels of antibiotics. The authors were seeking to understand more about how antibiotics given to humans and animals and subsequently released into wastewater and then receiving streams could impact the ecosystem-related functions of microorganisms in those streams. Laboratory microcosms were set up to expose the sediment to one of three antibiotics or to a mixture of the antibiotics. Nutrients also were added to the microcosms. The chemical and biological components of the microcosms were interrogated with numerous chemical and biological measurements over the course of a 7-day incubation. It was predicted that the antibiotics would have a significant reduction in the capability of the sediment community to conduct nitrogen transformations and uptake organic carbon. Ultimately the authors found that the antibiotics had fairly little impact on the functioning of these microbial communities. Nitrification may have been impacted during the oxic phase of the incubations (<24 hours) by the sulfonamide and mixed antibiotic treatments, but it was suggested further study would be needed to clarify this impact.

Overall, I found this to be a well conducted study and a cohesive manuscript. The results indicated that low levels of antibiotics over a short duration may not have a big impact on the nitrogen cycling or organic C uptake by the sediment microbial community in streams. This information will be useful in future work that addresses impacts of additional antibiotics/pharmaceuticals or varying concentrations of these substances on aquatic microorganisms and/or the ecosystem services provided by these systems. I do not have any major criticisms of the work, but I do have some minor points that need clarification and questions about some of the results interpretation presented in the discussion.

Also, please see the details regarding PLOS ONE data availability. A researchgate account is listed as the place where data is housed, but I did not see the underlying data upon going to that account. PLOS asks that all data used to calculate means, etc. are available in their raw count form. Please make these data available as supplemental info or in another data repository

tory.

Authors general comments: Reviewers 1 & 2 each made comments specific to grammatical issues throughout the text. Those comments can be found below. We have gone through the document and made necessary changes to the text in reflection to their comments. Other edits can be found in the text and marked via track changes. Comments specific to grammar have been included below. Reviewers also commented on public data sharing. Raw data for this project can be found at https://doi.org/10.6084/m9.figshare.17008964.v1
https://doi.org/10.6084/m9.figshare.17009126.v1

Reviewer 1

Title: “impacted” instead of “impact”

L85: “irreversibly bind”

L87: “is a synthetic”

L89: delete “enzyme of”; “Prior environmental studies”

L115: “In the laboratory, ash-free dry mass”

L143: is this the correct formatting for Figure reference?

L179: MIMS

L227: “spiked”

L296: note in text “copies per gram sediment”

L305-306: the wording on this statement needs to be fixed, and clarified

L327: “in” instead of “from”.

L344 and L345: Sentences should not begin with “This”: specify what the subject is.

Changes to the text for this specific comment are reflected in the revised manuscript (L344-345)

L335 and L337: Start these sentences with Underwood et al. [26] instead of a number

Supplemental Table 3: There is a lingering comment in the document, please fix.

Reviewer 2

L85 – should be “binds” not “binding”

L95 – should be “common” not “commonly”

L227 – should be “spiked” not “spike”

L232 – delete “used”

L305 – should be “there” instead of “the”

L191 – This is semantics, but by convention it is written as the “16S rRNA gene” and is not italicized. Italics are used for protein-coding genes.

Reviewer 1 comments to address

Reviewer comment: L108 – What is meant by “for the assay” in this sentence? Is this the water that was collected for all the experimental setups? Please clarify.

Response: Thank you for commenting on this. Yes, the sediment and surface water that was collected from this forested stream were used in microcosm construction. We have revised the text to reflect these changes (L109)

Reviewer comment: L109-110 – Please cite this work, if possible, even if it is from a non-peer reviewed source. It would be useful to have the data showing that antibiotics are undetectable at this site – then the reader could evaluate what antibiotics were assayed and what method was used, etc.

Response: We regret to inform the reviewer that the data was not previously available in a source that is referenceable. To provide transparency we added the chromatograph to our open source repository at https://doi.org/10.6084/m9.figshare.17009126.v1 . Antibiotics were identified initially based on the [M+H] + ion m/z signature. In using MZmine software we found no ion masses or chemical structures that were reflective of the antibiotics utilized in the present study as well as those detailed in:

• Pugajeva, I., Rusko, J., Perkons, I., Lundanes, E., & Bartkevics, V. (2017). Determination of pharmaceutical residues in wastewater using high performance liquid chromatography coupled to quadrupole-Orbitrap mass spectrometry. Journal of pharmaceutical and biomedical analysis, 133, 64-74.

Reviewer comment: L238 and all Figures: There are no post-hoc letters on the figures in my downloaded pdf. Please make sure these come through; it’s harder to interpret without them.

Response: We have adjusted the figure to identify differences through specific details. As stated in the figure legends, the posthoc letters are only for the first figure rate graphs. Thus, you can find the letters only in figure since there were no other significant differences in other figures.

Reviewer comment: L327-332: The consideration of additive vs. synergistic effects in the discussion was intriguing. Please revisit that here. I expect the authors don’t want to get too speculative but there is an observation about reduced uptake vs. cell death that is possible. It would be even more interesting to do a back-of the envelope calculation of how many bacteria would have died to produce this much ammonium over a day….

Response: We love this suggestion and the reviewers enthusiasm for this really interesting problem of additive vs. synergistic effects. While we would love the freedom to speculate more wildly, our results just aren't giving enough insight to feel comfortable pushing this idea in the discussion. It was a major motivation of the entire study, but the effects of our treatments were too subtle to make any strong inferences. That's good news for sediment microbes, but doesn't push the mechanistic understanding as far as we would like. This will definitely be motivating future and more elegant experimental designs.

Reviewer comment: L341: sorption of organic compounds is likely a major mechanism underlying apparent resistance or resilience, please add more emphasis and consideration here. Is the resistance/resilience biological, or only apparent due to physical buffering – would there be a threshold where chronic effects manifest?

Response: Thank you for pointing this out as a topic that should be included. Sorption does play a role in the biological inactivity or activity of xenobiotics. However, we felt that this mechanism may not be as influential due to the percent organic matter for sediment being relatively low at 1.2% and sediment particles being more composed to course particles rather than fine, which would alter sorption as well. We did include information in the revision pertaining to the physicochemical properties of antibiotics and how their interaction with sediment may favor sorption or desorption which may influence their biological activity in sediments and our results. Changes to the text can be found on L342.

We also included information on the results also being attributed to the MIC for each antibiotic used (0.25 to 16 ug/ml or 250 to 16000 ug/L), thus, indicating that levels used may not have been high enough to promote a biological response. 

Reviewer comment: L344: please don’t imply that evolutionary processes were happening over a day or a week (unless that is what is meant and if so needs to be substantiated); delete “adapting to or”, replace with “acclimating” if desired

Response: We appreciate this comment and suggestion. We have revised the text to not include misleading statements. 

Reviewer comment: L344: Re lack of differences: are sulfamethoxazole, danofloxacin, and erythromycin anaerobe-effective? Also, antibiotic resistance exists in plenty of populations even in “naïve” sediments: many bacteria compete with one another, like all organisms.

Response: We appreciate this comment, previous work (referenced below) has shown that anaerobic bacteria are susceptible to antibiotics used in the study. 

• SMX (Wüst, J., & Wilkins, T. D. (1978). Susceptibility of anaerobic bacteria to sulfamethoxazole/trimethoprim and routine susceptibility testing. Antimicrobial agents and chemotherapy, 14(3), 384-390.) 

• ETM (Watt, B. (1977). Erythromycin and anaerobes: in vitro aspects. Scottish medical journal, 22(1_suppl), 389-391.), and for 

• DAN (Papich, M. G., & Watts, J. L. (2017). New interpretive criteria for danofloxacin antibacterial susceptibility testing against Mannheimia haemolytica and Pasteurella multocida associated with bovine respiratory disease. Journal of veterinary diagnostic investigation, 29(2), 224-227. 

Reviewer comment: L346-348: This is true but it would be more interesting to elaborate on the cellular mechanism: Unless cells are killed and the DNA decomposed, there will be no decline in gene copies; but in the short-term cellular functions including reproduction should be inhibited. In fact, the same would apply to metagenomic assessment so I don’t agree with the viability of metagenomics as a next step as posed in L349. Considering which bacterial populations are fastest-growing and which biogeochemical functions they support might be more interesting in this paragraph.

Response: We appreciate the reviewers’ comments and have put more emphasis in to follow-up analysis that would benefit these types of studies, particularly moving from metagenomics to metatranscriptomics to create a functional profile of genes expressed that are specific to biogeochemical processes. Revised text for this can be found on L373-378.

Reviewer comment: L373-374: This sentence could benefit from some grammatical attention (who are “they”- and there are two different implied subjects, one in each question listed, neither subject is stated directly) and also some elaboration on what is meant by “cost”. Please improve this statement.

Response: Thank you for your comment, we initially were trying to convey that although microbes seem to be resistant to pressure from antibiotics, there may be other measurable endpoints that are impacted that scientist have not accounted for, warranting future work to understand more how mixtures effect these communities. We have made edits to that section of the discussion to provide clarity, improved grammar, and a more thoughtful conclusion.

Reviewer comment: Authors are required to make raw data publicly available, if I understand correctly, and I do not see a link to data repository in the manuscript.

Response: We have uploaded the raw data and additional tables to figshare (https://figshare.com/articles/dataset/_/17008964). 

Reviewer 2 Comments to Address

Reviewer comment: L312 – I agree that antibiotic mixtures are a more realistic scenario experienced by aquatic microbes, but how are the mechanisms behind their impact more complex than a single antibiotic exposure? Is there evidence that mixtures create a larger impact than would be predicted based on responses to individual antibiotics? Further support for this statement is needed or it needs to be presented as an opinion.

Response: Thank you for your comment, we have made revisions in the document to reflect these changes. The rationale for the text is due to the potential for mixtures to act in ways that differs from single exposure. While the references in the text are specific to cyanobacteria, we believe it is still relative to the overall subject of mixture toxicity vs single. Revised changes can be found on L312-L315

Reviewer comment: L314 – 318 – Could another alternative be that the concentration of antibiotic added was not at a level that would impact many microbes? What is the minimum inhibitory concentration for these antibiotics and how does this relate to the concentration used? Have other observed effects on microbial activity at these concentrations. If not, then it is possible that the dose was too low to impact process. I agree that low concentrations create a more realistic scenario, and the ones used here matched measured levels in other waterways, so it was a good target.

Response: Another reviewer mentioned this as well. We have added in additional text, references, and elaboration in the discussion to address this (L353-357). 

Reviewer comment: Perhaps a more continuous dosing of antibiotics, which could be brought by wastewater discharge would have a larger effect at this low concentration than a single pulse of the antibiotic. I believe additional discussion related to the antibiotic concentration used would aid in interpretation of the results.

Response: Thank you, we believe that is a great idea. We added additional text specific to follow up studies and utilizing pulses rather than continuous exposure, which mimic antibiotic occurrence in the natural environment, specifically urban streams that receive effluent. Revised text can be found at (L389-392). 

Reviewer comment: Also in this section – in what evidence is there that these communities are highly resilient? My question isn’t whether the communities are resilient – this is a hypothesis, but more how are the authors defining that term. I tend to define it as the ability to recover from disturbance to a pre-disturbance state. This would indicate that at some point there was a big change in microbial community following antibiotic addition, but it recovered quickly. To me that data don’t support that as being a primary conclusion. I see it more that the community was resistant to this disturbance – i.e. they were not impacted or that there was enough redundancy in the community that it superseded any losses. A more specific indication of what is meant here by resilient would clarify what hypotheses the authors are considering in the interpretation of their data. L317 – 318 – Why not discuss this here then? What do you think it means? Or what do you mean exactly by this statement?

Response: Thank you for a thoughtful comment, we have taken this into consideration and have removed text in this area and revised this topic so that is later discussed in the manuscript on (L353-363)

Reviewer comment: L323 – In what way does being a broad or narrow spectrum antibiotic matter to nitrification? Be more specific in this statement.

Response: We have taken this comment into consideration and added additional text to clarify the intent. In short, narrow spectrum targets a specific group of bacteria, while broad targets both gram negative and positive bacteria. Nitrifiers are typically gram positive and each antibiotic used was a broad spectrum, demonstrating that although they may be classified as broad spectrum, their impact on processes such as nitrification can vary (L323-327). 

Reviewer comment: L340 - 342 – Clarify this. I think I agree with your point but make clear the distinction between the studies. Did the previous study also use sediment? Sediment inclusion could make a big difference as diffusion of the antibiotic into the sediment may not be that high in 7 days.

Response: Thank you, for clarification that study referenced did not use sediment. We have reflected this change in the revised text (L341-351)

Reviewer comment: The wording, “…antibiotics exposed within sediment…” is a bit confusing. Is this stating that the antibiotics were added to the sediment or that the microbes in the sediment rather than the water were exposed to the antibiotics?

Response: We have revised the text in the section to clarify our rationale and line of thought. (L345)

Reviewer comment: L343 – More explanation is needed in this paragraph? I do not follow the logic that the communities changed in structure, but this led to a lack of difference in measured gene copies. If the community composition is changing rapidly, I would expect the nitrogen-related gene concentrations to change rapidly as well, as only a few select taxa are capable of many of the relevant processes (L355-360).

Response: We agree with the reviewer that our initial version included logic that did not tie in together with our findings. We revised the text to include more information on MICs of antibiotics and the possibility of concentrations being too low to observe an effect along with natural resistance that microbial communities develop even when not exposed to antibiotics that may have negated any effects. 

Reviewer comment: I agree with the reference cited that the concentration of a particular gene often does not relate to the activity levels at any given moment for that gene product. To get at this idea, gene expression (mRNA or protein copies) would need to be quantified. However, here the end products (N2 gas, etc.) were measured and they were similar between the control and treatment, so I would not expect that gene expression levels changed much either. I would suggest metatranscriptomics rather than metagenomics would be a better measure – find out which organisms are actually responding to the different microcosm setups.

Response: We have looked into this more and added specific text explaining why metagenomic may be limited in these types of studies and how metatranscriptomics would benefit future studies. Revised text can be found at (L373-378)

Reviewer comment: L373 – 374 – The wording of this sentence is not clear. Please re-phrase. The negative, “…not how they are impacted…” make it difficult to interpret. Also, what is meant by “cost”? Is there a cost to the microbes for resisting antibiotics?

Response: This was also addressed in Reviewer 1 comments and the closing of the discussion has been revised to remove any text that was previously hard to interpret.

---

## [Editor Report · Decision Letter 1]

9 Dec 2021

Are Nitrogen and Carbon Cycle Processes Impacted by Common Stream Antibiotics? A Comparative Assessment of Single vs. Mixture Exposures

PONE-D-21-30040R1

Dear Dr. Gray,

We’re pleased to inform you that your manuscript has been judged scientifically suitable for publication and will be formally accepted for publication once it meets all outstanding technical requirements.

Kind regards,

John J. Kelly

Academic Editor

PLOS ONE
---

## [Editor Report · Acceptance letter]

14 Dec 2021

PONE-D-21-30040R1 

Are nitrogen and carbon cycle processes impacted by common stream antibiotics? A comparative assessment of single vs. mixture exposures 

Dear Dr. Gray:

I'm pleased to inform you that your manuscript has been deemed suitable for publication in PLOS ONE. Congratulations! Your manuscript is now with our production department. 

Kind regards, 

on behalf of

Dr. John J. Kelly 

Academic Editor

PLOS ONE